# High Current Field Emission from Large-Area Indium Doped ZnO Nanowire Field Emitter Arrays for Flat-Panel X-ray Source Application

**DOI:** 10.3390/nano11010240

**Published:** 2021-01-18

**Authors:** Yangyang Zhao, Yicong Chen, Guofu Zhang, Runze Zhan, Juncong She, Shaozhi Deng, Jun Chen

**Affiliations:** Key Laboratory of Optoelectronic Materials and Technologies, Guangdong Province Key Laboratory of Display Material and Technology, School of Electronics and Information Technology, Sun Yat-Sen University, Guangzhou 510275, China; zhaoyy53@mail2.sysu.edu.cn (Y.Z.); chenyc25@mail.sysu.edu.cn (Y.C.); zgfbb@126.com (G.Z.); zhanrz3@mail.sysu.edu.cn (R.Z.); shejc@mail.sysu.edu.cn (J.S.); stsdsz@mail.sysu.edu.cn (S.D.)

**Keywords:** Indium doped ZnO nanowire, thermal oxidation, field emission, flat-panel X-ray source

## Abstract

Large-area zinc oxide (ZnO) nanowire arrays have important applications in flat-panel X-ray sources and detectors. Doping is an effective way to enhance the emission current by changing the nanowire conductivity and the lattice structure. In this paper, large-area indium-doped ZnO nanowire arrays were prepared on indium-tin-oxide-coated glass substrates by the thermal oxidation method. Doping with indium concentrations up to 1 at% was achieved by directly oxidizing the In-Zn alloy thin film. The growth process was subsequently explained using a self-catalytic vapor-liquid-solid growth mechanism. The field emission measurements show that a high emission current of ~20 mA could be obtained from large-area In-doped sample with a 4.8 × 4.8 cm^2^ area. This high emission current was attributed to the high crystallinity and conductivity change induced by the indium dopants. Furthermore, the application of these In-doped ZnO nanowire arrays in a flat-panel X-ray source was realized and distinct X-ray imaging was demonstrated.

## 1. Introduction

Larger area one-dimensional (1D) cold cathode arrays play an important role in panel light sources, flat-panel X-ray sources, and photodetectors [1,2,3,4]. In these materials, field emission properties from zinc oxide (ZnO) nanowire has been extensively studied in recent years [5,6,7,8,9,10,11], demonstrating facile synthesis of uniform large-area samples that possess low turn-on fields, good emission stabilities, and uniform emission site [12,13]. Therefore, large-area ZnO nanowire field emission arrays (FEAs) have been studied and their subsequent application in various vacuum microelectronic devices has been demonstrated. Chen et al. have reported a larger-area flat-panel X-ray source comprising a ZnO nanowire field emitter that produced a uniform distribution of X-ray generation owing to the uniform electron emission [2]. Furthermore, a double-sided radiating large-area flat-panel X-ray source device has been reported that produced clear X-ray images with high contrast [14]. Zhang et al. have prepared a flat-panel photodetector using ZnO nanowire FEAs exhibiting an excellent light response performance in a wide spectral range [15].

For these applications, a high emission current is essential to obtain a high-performance device. For example, a high emission current would make possible a high-brightness flat-panel X-ray source and thus a short exposure would be adequate for imaging. Whereby, high-speed X-ray imaging of moving objects could be realized without blurring. Previous studies have explored various approaches to enhance the ZnO nanowires emission current, including changing the morphology to increase the field enhancement factor, reducing the screening effect by optimizing the population density, lowering the work function by post-treatment, and improving the electrical property by lowering the back contact or via nanowire element doping [7,16,17,18,19]. Among these methods, element doping is an effective way to enhance the emission current by changing the nanowire conductivity and the lattice structure [20,21]. Zinc oxide is a typical n-type doped semiconductor owing to the intrinsic oxygen vacancies, and n-type doping is typically preferred to reduce the resistivity of ZnO nanowires. In previous studies, the most commonly used metal element dopants are Al, Ga, In, and Ge [20,21,22,23,24,25,26,27].

Various attempts to prepare doped ZnO nanowires have be made including both hydrothermal growth [21,28] and thermal evaporation [23,24]. Lv et al. have synthesized Al-doped ZnO nanowires by a facile hydrothermal method under a 100 °C solution [21]. Ahmad et al. used thermal evaporation to prepare In-doped ZnO nanowires at 700 °C and 133.32 Pa pressure, in which the Zn, ZnO and indium powders was mixture as the ingredient [24]. In addition, the growth of ZnO nanowires with Cu, Ga, and Sb elements doping has been reported using the thermal evaporation method with typical growth temperatures over 700 °C [29,30,31]. Most studies using thermal evaporation have required a high temperature to achieve high-quality nanowire growth, which precludes the use of the glass substrates essential for large-area sample preparation.

Previous researches have shown that the influence of element doping on the electrical and field emission performance of ZnO is significant. Nunes et al. have reported the effects on the electrical properties of ZnO films with the Al, Ga, and In element doping, reporting a lowest resistivity of ZnO film with 1 at% of indium doping [32]. Zhou et al. have synthesized ZnO nanowires with Ga doping using chemical vapor deposition method, where the nanowire resistivity decreased by two orders of magnitude (10^−1^ to 10^−3^ Ω·cm) after doping [25]. In addition to these elements, researchers have also studied the effects that doping ZnO nanowires with other elements on its electrical properties. Improved ZnO nanowire electrical properties were observed with F-doping by post-processing [33]. Ahmad et al. have synthesized ZnO nanowires with In doping that exhibited a superior field emission performance [24]. These previous results demonstrated that element doping can effectively improve the electrical properties of ZnO nanowires and thereby enhance their field emission properties.

Thermal oxidation is a feasible method to prepare ZnO nanowire FEAs on large-area glass substrates owing to its low temperature growth [6]. Furthermore, because the ZnO nanowires are directly grown from Zn films, the nanowires can be easily patterned by a microfabrication process thereby easily achieving integration of nanowires in gated FEAs [15,34,35,36,37]. Tuning the nanowire field emission properties has been attempted using element doping. For example, element doping via plasma post-treatments in a NH_3_ plasma or via low-energy ion implantation with fluorine doping has been reported [33,38]. However, direct element doping in ZnO nanowires during the thermal oxidation preparation process has not yet been realized, primarily owing to the relatively low growth temperature. However, the melting point of indium matches the thermal oxidation growth temperature, and thus it is feasible to grow In-doped ZnO nanowires directly with indium as a doping element using thermal oxidation. However, the attempt has not yet been reported.

In this study, we synthesized large-area In-doped ZnO nanowire arrays on an indium-tin-oxide (ITO)-coated glass substrate by the thermal oxidation method. The structure and optical properties of the nanowires were characterized, and a self-catalytic thermal oxidation growth method was proposed to explain the growth of the ZnO nanowire with indium doping. The field emission properties and maximum current of the prepared In-doped nanowires were measured, confirming the excellent performance of the samples. Furthermore, a flat-panel X-ray source was fabricated using the ZnO nanowire array with indium doping as the cathode, which achieved uniform emission and clear X-ray imaging of biological and non-biological samples. These results are of great significance to the application of large area indium doped ZnO nanowires in vacuum electronic devices.

## 2. Experimental

ZnO nanowires with indium doping were prepared via thermal oxidation of an In-Zn alloy thin film. Arrays of round patterns were prepared on glass substrates, where each round pattern had a 5 μm diameter and 1600 × 1600 patterns existed on a 4.8 × 4.8 cm^2^ area with a 25 μm spacing between adjacent patterns. The pattern diameter and spacing were chosen to minimize the screening effect, according to a previous work [19]. To observe the cross-sectional morphology of the nanowires, the ZnO nanowires were also prepared on silicon substrates.

The sample preparation process started with depositing an ITO film 520 nm thick on a glass substrate, where the ITO film acted as the electrode layer. Patterns of photoresist were formed on the ITO electrode layer by ultraviolet (UV) lithography. Bombarded with Ar ions for 30 min prior to deposition of the indium-zinc (In-Zn) alloy film on the pattern via electron beam evaporation. The evaporation source was the In-Zn alloy particles comprised 20% In and 80% Zn (Aoshi Science and Technology Co. Ltd., Shenyang, China). The In-Zn alloy film thickness was about 2 μm. whose pattern was formed using a lift-off method. Finally, the sample was grown by thermal oxidation in which placed in a three-temperature zone tubular furnace. During growth, the temperature was increased from room temperature to 470 °C in 192 min, and maintained at this temperature for 3 h in ambient atmosphere, and then cooled to room temperature naturally. Through this process, indium doped zinc oxide nanowires arrays were obtained.

The morphology and structure of the prepared In-doped ZnO nanowires were characterized by scanning electron microscopy (SEM; SUPRA™60, Zeiss, Oberkochen, Germany). The crystalline structure was examined by X-ray diffractometry (XRD; D/Max-IIIA, Rigaku Corporation, Tokyo, Japan) using Cu K*α* radiation (1.54056 Å wavelength) and by high-resolution transmission electron microscopy (HRTEM; Titan G2 300 KV, FEI, Columbus, NJ, USA). Also, energy-dispersive X-ray spectroscopy (EDX) in the HRTEM instrument was used to verify to the In doping and to visualize the distribution of the In atoms via element mapping. Further determination of the chemical composition was accomplished with X-ray photoelectron spectroscopy (XPS; ESCALAB 250 system, ThermoFisher, Waltham, MA, USA) using Al Kα X-rays (1486.6 eV, 150 W), in which the energy scale of the spectrometer was calibrated with the C1s peak (284.8 eV). Ultraviolet photoelectron spectroscopy (UPS; ESCALAB 250 system, ThermoFisher, MA, USA) was used to measure the sample work function. In addition, Raman spectra were obtained by Raman spectroscopy (FLSP920, Edinburgh Instruments, UK.) with a 523 nm wavelength Ar ion laser source. Photoluminescence (PL) of single In-doped ZnO nanowires was measured using the Raman spectroscope (FLSP920, Edinburgh Instruments, UK) with a 325 nm wavelength UV light from a He-Cd laser at room temperature. The sample was prepared by dripping dispersed nanowire solution on a clean silicon wafer prepared by scraping nanowires off the substrate and ultrasonically dispersing them in alcohol.

The field emission properties were measured in a vacuum chamber with a base pressure of 2.0 × 10^−5^ Pa. The current-voltage (I–V) characteristics were measured using a diode structure from which, after going through an aging process, the field emission current versus applied voltage properties were recorded. To observe the emission uniformity, a phosphor screen was used as the anode upon which the field emission image was recorded using a digital camera. The gap between the phosphor screen and cathode was 0.25 mm. To measure the maximum field emission current, a quartz glass substrate coated with an ITO layer was used as the anode, where the gap between the anode and cathode was 0.12 mm.

A diode-structure flat-panel X-ray source was fabricated with the In-doped ZnO nanowire array as the cathode and a 1 μm thick molybdenum thin film on silica glass as the anode. The gap between the cathode and anode was 6 mm, and the device was placed in a vacuum chamber measuring 1.0 × 10^−6^ Pa. The energy spectra of the X-rays generated by the planar source were recorded by an X-ray detector (X-123SDD, AMPTEK, Inc, Bedford, MA, USA), while an X-ray dosimeter (Magic Max, IBA, Göttingen, Germany) was employed to measure the radiation dose. Contact X-ray images were captured by a dental digital imaging detector (RSV4, Visiodent, Clichy, French) for small objects and a flat-panel imaging detector (Xineos-1515, Thousand Oaks, CA, USA) for large objects.

## 3. Results and Discussion

### 3.1. Morphology and Structure Characterization

Figure 1a shows a photograph of the In-doped ZnO nanowire FEAs prepared on ITO/glass substrate. A low-resolution SEM image featuring a smaller 4 × 3 section of the pattern array is shown in Figure 1b, while Figure 1c_1_,c_2_ exhibit typical high-resolution SEM images of a single pattern in the region of (a_1_), (a_2_) shown in Figure 1a. Figure 1d shows the typical cross-sectional SEM image of a single pattern. From the figure, we can see the In-doped ZnO nanowire have a tapered morphology near the bottom and a nanowire morphology at the top. Furthermore, we statistically summarize the lengths and diameters of about 100 nanowires distributed across different patterns. The distribution is shown in Figure 1e,f. From the figure, we can see that most nanowires have lengths between 3.5 μm and 4 μm and diameter of about 70 nm. These results show that a uniform growth of nanowires was obtained on the substrate. The nanowire population density is estimated to be about 9 × 10^8^ cm^−2^ by a statistical analysis.

Figure 2a shows a TEM image of a single typical nanowire and its corresponding EDX spectrum (inset in Figure 2a). The EDX spectrum exhibits an obvious indium peak demonstrating an approximate In dopant concentration of 1 at%, proving the presence of indium in the nanowires. Furthermore, a statistical analysis of doping concentration in five individual nanowires shows the doping concentration only has a small fluctuation and the average concentration is around 1%. Figure 2b shows a HRTEM characterization of the single nanowire in Figure 2a and the corresponding selected-area electron diffraction (SAED) pattern (obtained from marked region in Figure 2a). The lattice fringe spacing between adjacent lattice planes in Figure 2b is measured as 0.29 nm and the SAED pattern indicates that the nanowire has good crystal quality (inset in Figure 2b). The EDX elemental mapping results for the Zn, O, and In elements are shown in Figure 2c_1_–c_3_, respectively, in which Zn and O exhibit a uniform distribution while In shows a random distribution in the nanowires. This result indicates that the In element was successfully doped into the nanowire.

Figure 3a shows the XRD characterize of the In-doped ZnO nanowire with the 2θ range 25°–60°, and from the result we can observe the peaks of ZnO (100), (002), and (101). Besides the peaks of ZnO, the peaks for In and Zn are also clearly presented, which originate from the residual In and Zn after nanowire growth in the thin film underneath the nanowires. The XPS spectrum of the prepared sample is shown in Figure 3b and the binding energy with a range of 0-1200 eV. As shown in XPS spectrum, the Zn 2p_3/2_ and Zn 2p_1/2_ peaks located at ~1017.4 and ~1040.39 eV are observed with the energy difference of 22.99 eV. In addition, the inset of Figure 3b shows the enlarged peaks for In 3d_3/2_ and In 3d_5/2_ which located at 445.3 and 452.5 eV and the energy difference is 6.8 eV. Closer inspection of the In-doped ZnO nanowires shows that, compared with the standard values of In (7.5 eV) and Zn (22.97 eV) [39,40], the In 3d peaks exhibit a positive shift which is probably caused by electron transfer from ZnO to In due to the strong electronic interaction between In and oxide support, which agrees with early report [24]. Meanwhile, the Zn 2p peaks exhibit a negative shift due to the electronegativity difference between Zn and In [24]. The O1s spectrum at 530.3 eV is also observed in the XPS scan. From the XPS results, the In content can be calculated and the value is about 15%. The result is much higher than that obtained from the in-situ EDX analysis in TEM. We think the high In content obtained from XPS originates from the thin film under the nanowires. After growth, the In-Zn film was partially oxidized and there are residual In and Zn in the ZnO thin film, which is confirmed by the XRD result. In the XPS measurement, the signal from the residual In element in the ZnO film will mix with the signal from the nanowires. Thus, a high In concentration was obtained in the XPS results. Therefore, the result from TEM analysis is more reflecting the true composition of the nanowire.

Figure 3c shows the normalized PL spectra of a single ZnO nanowire with indium doping, where the one observed UV emission peak around 390 nm corresponds to the intrinsic band edge emission. This band edge emission peak of the ZnO nanowire with indium doping is slightly blue-shifted compared with the previous results of an without element doping ZnO nanowire (UV band peak at 394 nm) [27]. That is attributed to the shift of optic band gap owing to indium dopants contribute an increased number of electrons [24]. In addition, some early studies have reported two emission bands typically observed in the PL spectrum of ZnO nanowires, where the additional peak around 550 nm is owing to the existence of oxygen vacancies [19,27]. However, in this study this additional peak at the visible range was not observed, which implies a good crystal quality of the In-doped ZnO nanowires consistent with the TEM observations. In addition, the previously-reported without element doping ZnO nanowires grown by the thermal oxidation method exhibited strong green (~550 nm wavelength) PL peak corresponding to the oxygen vacancies [16,19]. The difference in the PL results of the two types of thermal oxidation-grown ZnO nanowires may indicate that different growth mechanisms may function in the with element doping and without element doping ZnO nanowires.

### 3.2. Growth Mechanism

Previous work has represented the growth mechanism of ZnO nanowires without element doping from a Zn film by thermal oxidation using a strain-induced mass diffusion model [7]. According to the model, the supply of Zn atoms from the Zn film promote the ZnO continuous growth. Therefore, a cavity structure within the ZnO film under the growing ZnO nanowires is observed after growth. After In-doped ZnO nanowire growth, however, the film morphology exhibits a dense layer, indicating that a different mechanism may exist. To investigate the growth mechanism of In-doped ZnO nanowires, we characterized the cross-sectional morphology of nanowires prepared on Si after various growth times. Figure 4a–d show the SEM images of In-Zn film surface morphology on Si substrates at the thermal oxidation times of 0 (20 °C), 90 (250 °C), 180 (470 °C), and 270 (470 °C) min, respectively, while Figure 4a_1_–d_1_ respectively show corresponding cross-sectional SEM images (where the inset of Figure 4c_1_ is a magnified image). The trend of film thickness with heating time is plotted in Figure 4e.

As can be seen from Figure 4a, the prepared alloy film is a continuous film, and the cross-sectional SEM image shows a discrete columnar structure 1.2 μm in thickness. After heating for 90 min, the film is melted and some small protrusions are visible on the surface of the film, which thins to 1 μm thickness (Figure 4b,b_1_). With increased heating time to 180 min and a temperature reaching 470 °C, significant film melting occurs, larger protrusions appear on the surface, and a dense film is formed (Figure 4c,c_1_). At the same time, some short nanowires emerge (inset of Figure 4c_1_) and the film thickness reaches ~900 nm after 180 min. Under the temperature at 470 °C for 90 min, the film thins and the nanowires lengthen. In addition, the film exhibits a rugged surface (Figure 4d,d_1_) and is about 700 nm in thickness.

Analyzing the phenomena revealed by these SEM images, a model was proposed to describe the growth process of the ZnO nanowires with indium doping, which is illustrated in Figure 5. Previous studies have shown that element doped ZnO nanowire could be synthesized by a vapor-liquid-solid (VLS) mechanism [23,24,29,31]. We follow a self-catalytic VLS growth mechanism to explain the growth of the In-doped ZnO nanowires herein. When the In-Zn alloy film was placed in the quartz tube furnace and the temperature rises (Figure 5a), the relatively low melting point of indium cause indium agglomerates to precipitate on the film surface, which can act as catalysts (Figure 5b). Further increasing the temperature causes the zinc vapor, catalyzed by the indium agglomerates, to react with oxygen atoms in the air (Figure 5c) to form ZnO nanowires (Figure 5d). At the growth temperature of 470 °C, the vapor pressure of Zn is about 100 Pa [41]. Furthermore, during growth process, the zinc in the film is oxidized into zinc oxide, which is an exothermic process. The released heat will raise the local temperature, which might lead to an even higher Zn vapor pressure during the growth. Simultaneously, some indium atoms diffuse into the growing ZnO nanowires, thus forming In-doped ZnO nanowires. With increased growth duration, the indium particles become smaller and eventually are depleted. Therefore, the nanowires exhibit a morphology that is narrow at the top and wide at the bottom (Figure 5e). In addition, the vaporization of zinc reduces the Zn film thickness (Figure 5c–e).

It is worthy to note the growth mechanism is different from that for the ZnO nanowire grown from pure Zn film using thermal oxidation method. In that case, the as-grown ZnO nanowires have a twin-crystalline structure with growth direction of 110 and high oxygen vacancies existed in the nanowire. While in this work, highly crystallized ZnO nanowires were obtained.

### 3.3. Field Emission Properties

The field emission properties of a 4.8 × 4.8 cm^2^ In-doped ZnO nanowire array were studied. Figure 6a shows the curve the field emission current density versus the applied electrical field (J-E) and the inset shows the corresponding Fowler-Nordheim (F-N) plot of the ZnO nanowires with indium doping. The results indicate a turn-on field of the prepared sample (corresponding to current density of 10 μA/cm^2^) is about 7.1 MV/m. The obtained maximum emission current reaches 20 mA, corresponding to a current density of 863.5 μA/cm^2^ at the electric field density 10.16 V/μm. The F-N plot shows non-linearity over the whole electric field range and could be divided into two linear sections at low electric fields and high electric fields, respectively (inset of Figure 6a). This nonlinearity may be caused by the heating effect and space charge effect [42], however the curve is bent downward at high field, so the space charge effect dominates. The field enhancement factor *β* is about 1560 was calculated with the F-N equation from the linear section of the F-N curve at low field finding [19]. In this calculation for *β*, the work function of 3.98 eV was used for the In-doped ZnO nanowires, which was measured using UPS.

Figure 6b,c show the field emission stability of current and voltage. The results obtained both under constant current mode and constant voltage mode were presented, which were obtained when the power supply operated in these two modes, respectively. Under constant current of 6.5 mA for 60 min the voltage value has a slight fluctuation from 1.08 to 1.03 kV; and at constant voltage of 1.0 kV for 60 min the emission current increases from 3.48 to 3.9 mA. The increase of emission current may be owing to the surface cleaning due to the heating effect induced by field emission current. To calculate the fluctuation of the emission current and applied voltage, we used [19]
(1)$$\delta_{current}=\frac{\sum_{i=1}^{n}\left|I_{i}-I_{average}\right|}{nI_{average}}\;\times \;100\%,$$(2)$$\delta_{voltage}=\frac{\sum_{i=1}^{n}\left|V_{i}-V_{average}\right|}{nV_{average}}\;\times \;100\%,$$
where *i* is for the point of time, and at some point in time the value of current is *I_i_* and voltage is *V_i_*; and *I_average_* is the average value of the current over the entire measurement period and *V_average_* is the same. The results show that the current fluctuation is 2.8% under a constant 1.0 kV voltage and the voltage fluctuation is 1.1% under a constant 6.5 mA current. These results indicate these ZnO nanowires with indium doping has an excellent field emission stability.

Figure 7 shows a comparison in the emission current and corresponding current density of nanowires reported in this paper and that reported in previous literatures [19,20,23,43,44,45,46,47,48]. We note that the current increases with the increase of emission area (Figure 7a); however, because the emission area increase is disproportionately larger than the emission current increase of the samples, the current density decreases with increasing emission area (Figure 7b). For the samples possessing a similar area [19], the emission current of the element doped samples is greater than that of the samples without element doping. Compared with the previous reports and for large-area samples, higher value of the ZnO nanowires with indium doping in this work are obtained. These results provide an important guarantee for the subsequent fabrication of large-area devices.

### 3.4. Electric Characteristics of Single Indium Doped ZnO Nanowire

In order to explore the cause of high emission current from indium doped ZnO nanowires, the electrical characteristics of single indium doped ZnO nanowire were measured using a probe technique. Typical electrical characteristic of an individual ZnO nanowire with indium doping was presented in Figure 8, inset depicts the corresponding SEM image during the measurement. We calculated the corresponding geometrical parameters, resistance and conductivity of eight ZnO nanowires with indium doping and listed in Table 1. The resistance of each In-doped ZnO nanowire was obtained by fitting each electrical I–V curve using the metal-nanowire-metal model reported in the literature [49]. The result shows that the conductivity of single ZnO nanowire with indium doping is in the order of 18-174 S/m. However, previous study showed the pure single crystal ZnO nanowires with wurtzite structure have lower conductivity about 10^−2^–10^−1^ S/m [26,50,51]. The conductivity of the indium doped ZnO nanowire reported in this work is about 3~5 orders of magnitude higher.

In conclusion, we believe the higher conductivity contributes to the obtained high emission. Also, we think good nanowire crystallinity is a crucial factor for the higher emission current of the ZnO nanowire with indium doping. Recent studied shows that field emission from ZnO follows a field-induced hot-electron emission and nanowires remain stable when heated above 900 K when operated under high current [52]. The TEM and PL results demonstrated that the In-doped ZnO nanowires are single-crystal structures with few defects, leading to the improvement in thermal stability of In-doped ZnO nanowires. Therefore, the In-doped ZnO nanowires could endure high temperature induced by the high current.

## 4. Application in Flat-Panel X-ray Source

A 4.8 × 4.8 cm^2^ flat-panel X-ray source was fabricated using the cathode discussed in Section 3.3. The X-ray source device comprised a diode structure whose schematic is shown in the inset of Figure 9a. A molybdenum thin film prepared on a quartz substrate was used as the anode, and the cathode and anode were separated by ceramic spacers with a height of 6 mm.

Figure 9a shows the I–V plot of the flat-panel X-ray source and the corresponding F-N plot, where the former demonstrates that the device has an emission current of 225 μA under a 46 kV anode voltage. The X-ray characteristics of this device was subsequently measured under this voltage. Figure 9b shows an image of the field emission under 46 kV anode voltage, where the bluish bright contrast corresponding to the field emission was owing to electron bombardment of the molybdenum-coated quartz glass. Repeated testing of the device over a few days demonstrated the repeatability of the results at high voltage. To analyze the uniformity of the emission as represented in the image of Figure 9b, we calculated the average emission area rate and found it to be 95%. We therefore conclude that the X-ray source exhibits a superior X-ray emission uniformity owing to the uniform emission of the In-doped ZnO nanowires.

Figure 9c shows the X-ray energy spectra of the device, which exhibits two characteristic peaks of molybdenum located near 17.5 and 19.6 keV. We note that the low-energy X-rays (i.e., <15 keV) are attenuated by the quartz glass. Furthermore, the change of X-ray dose rate with anode voltage was measured at the distances of 20 and 25 cm from the X-ray source, as shown in Figure 9d. At each distance, an increasing anode voltage resulted in an increased dose rate; while the dose rate decreased significantly with increasing the distance. A maximum radiation dose rate of 368 μGy/s was measured at a 44 kV anode voltage 20 cm from the front of the X-ray source.

The imaging characteristics of the X-ray source were tested by tape the non-biological and biological objects to the surface of the X-ray imaging detector. Figure 10a show the X-ray image of a microprocessor chip using the dental imaging detector placed 35 cm from the X-ray source, in which the X-ray image clearly shows the thin metal wire structure inside the chip. Figure 10b show the X-ray image of a calculator taken by the flat panel detector, with a 75 cm distance between the detector and the device. The internal structure of the calculator can be clearly resolved and the number of spring coils can be accurately counted. Meanwhile, biological images were also presented. Figure 10c show the imaging result of a hippocampal specimen which can clearly distinguish the skeleton joints and the head structure. Figure 10d show the imaging result of a fresh loach. From the image we can clearly distinguish its skeletal structure and the distribution of internal organs.

Furthermore, we image moving objects using the fabricated device when the device is driven by a pulsed high voltage supply. High voltage pulses of 46 kV with the widths of 250, 100, 50, 25, and 10 ms were applied to the device, respectively. A metronome was used as the imaging object where placed 55 cm from the device and the frequency of the metronome is set to 1 Hz. Figure 10e,f show the images of metronome needle obtained using pulse widths of 100 and 10 ms, respectively. We founded the continuous shadows appear on the image when the pulse width is 100 ms. When the pulse width is 10 ms, almost stationary image of the metronome needle could be captured. The results show that the fabricated device using ZnO nanowire with indium doping as cathode has potential application prospect in dynamic imaging.

## 5. Conclusions

ZnO nanowire arrays with indium doping were prepared by the thermal oxidation technique, where the In was successfully doped into the nanowires at a content of approximately 1at%. A high field emission current of 20 mA was acquired from the large-area (i.e., 4.8 × 4.8 cm^2^) In-doped ZnO nanowire FEAs owing to the good crystal quality of the In-doped ZnO nanowires. Excellent emission stability was also achieved under constant current and constant voltage. A large-area flat-panel X-ray source was fabricated using the ZnO nanowire array with indium doping as the cathode, where a radiation dose rate of 368 μGy/s was achieved. In addition, clear X-ray imaging was achieved both for static and moving objects. This work indicates that In-doped ZnO nanowires prepared by the thermal oxidation method are promising for applications in large-area vacuum microelectronics devices.

## Figures and Tables

**Figure 1 nanomaterials-11-00240-f001:**
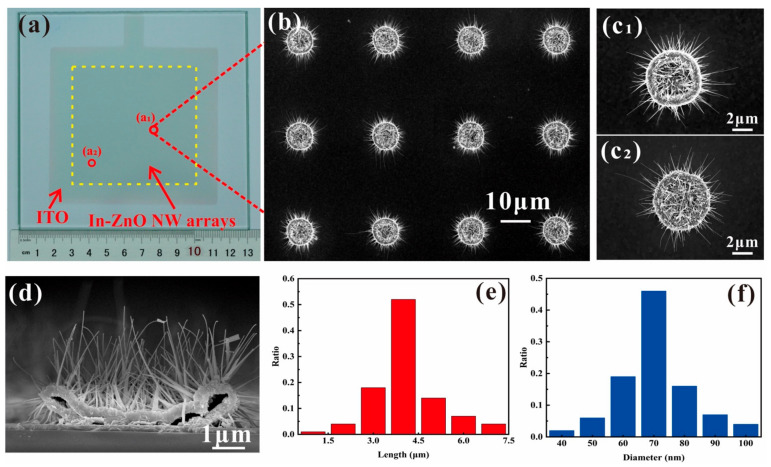
Images of ZnO nanowire arrays with indium doping. (**a**) Photograph consist of glass substrate, indium–tin-oxide (ITO) electrode and the prepared ZnO nanowire field emission arrays (FEAs) with indium doping; (**b**) top view SEM image of a 4 × 3 In-doped ZnO nanowire pattern array in low magnification; (**c_1_**,**c_2_**) single pattern at different regions in high magnification; and (**d**) cross-sectional SEM image of a single pattern; (**e**,**f**) Statistical results of the length and diameter.

**Figure 2 nanomaterials-11-00240-f002:**
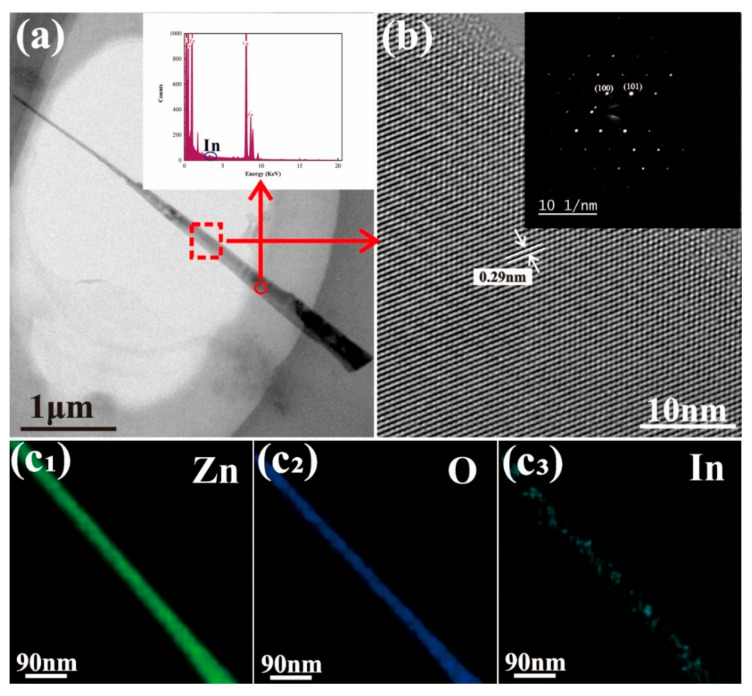
TEM characterization of a single In-doped ZnO nanowire. (**a**) Low-resolution TEM image; inset corresponding EDX spectrum. (**b**) High-resolution TEM image of the sample and selected-area electron diffraction (SAED) pattern (inset). (**c_1_**–**c_3_**) EDX elemental mapping of (**c_1_**) Zn, (**c_2_**) O, and (**c_3_**) In.

**Figure 3 nanomaterials-11-00240-f003:**
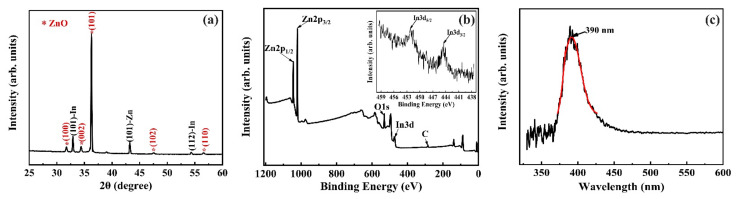
Characterization of ZnO nanowires with indium doping. (**a**) XRD pattern, (**b**) XPS spectrum, inset detail of XPS for In 3d enlarged peaks, (**c**) room-temperature PL spectrum of a single In-doped ZnO nanowire.

**Figure 4 nanomaterials-11-00240-f004:**
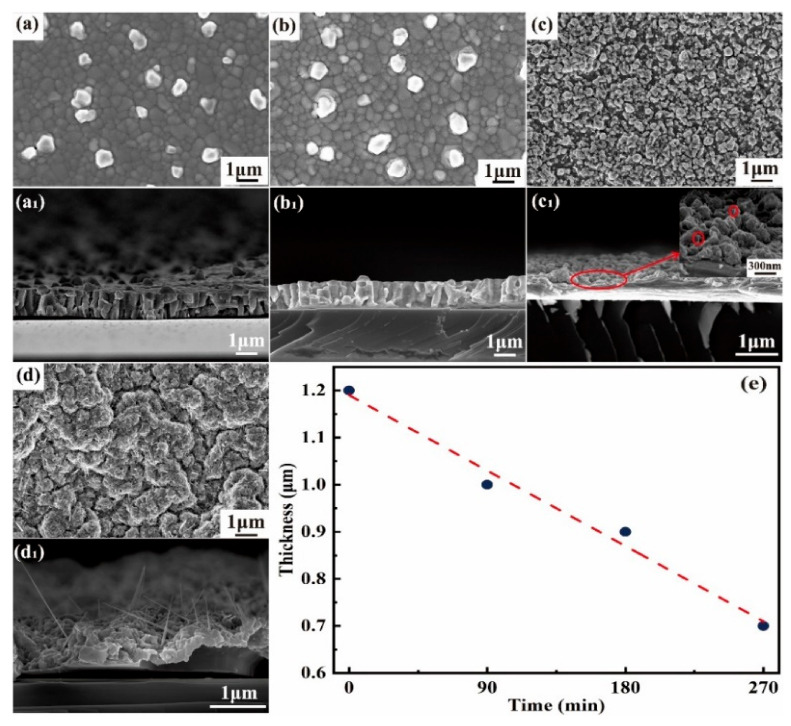
Top-view and cross-sectional SEM images of thin films and nanowires after a growth time of (**a**,**a_1_**) 0 min (20 °C), (**b**,**b_1_**) 90 min (250 °C), (**c**,**c_1_**) 180 min (470 °C), and (**d**,**d_1_**) 270 min (470 °C); (**e**) Zn film thickness vs. heated time.

**Figure 5 nanomaterials-11-00240-f005:**
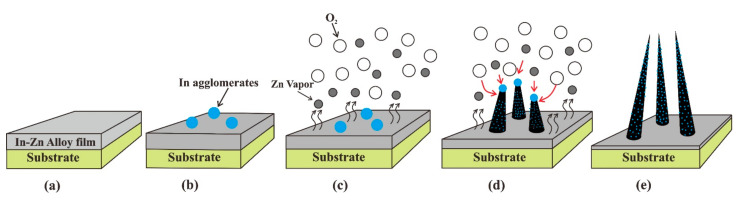
Schematic diagram of the growth process of In-doped ZnO nanowires. (**a**) The In-Zn alloy film was deposited on the substrate, (**b**) indium agglomerates precipitated on the film surface, (**c**) zinc vaporized from the film, (**d**) ZnO nanowires grew from the In catalyst, (**e**) the In-doped ZnO nanowire after growth.

**Figure 6 nanomaterials-11-00240-f006:**
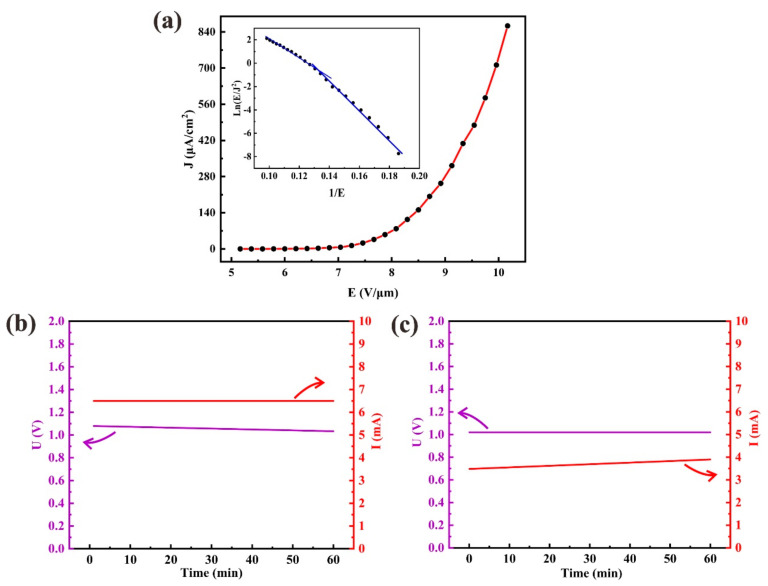
(**a**) Field emission curve (J-E) and Fowler-Nordheim (F-N) plot (inset) of ZnO nanowires with indium doping (**b**,**c**) Field emission stability of (**b**) voltage (U) with constant current at 6.5 mA and (**c**) direct current (I) with constant voltage of 1.02 kV.

**Figure 7 nanomaterials-11-00240-f007:**
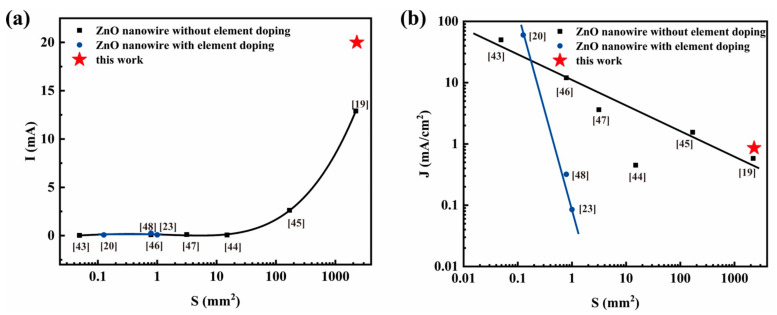
Comparison in the (**a**) emission current and (**b**) corresponding current density of nanowires reported in this paper and that reported in the literature (References given with data in square brackets).

**Figure 8 nanomaterials-11-00240-f008:**
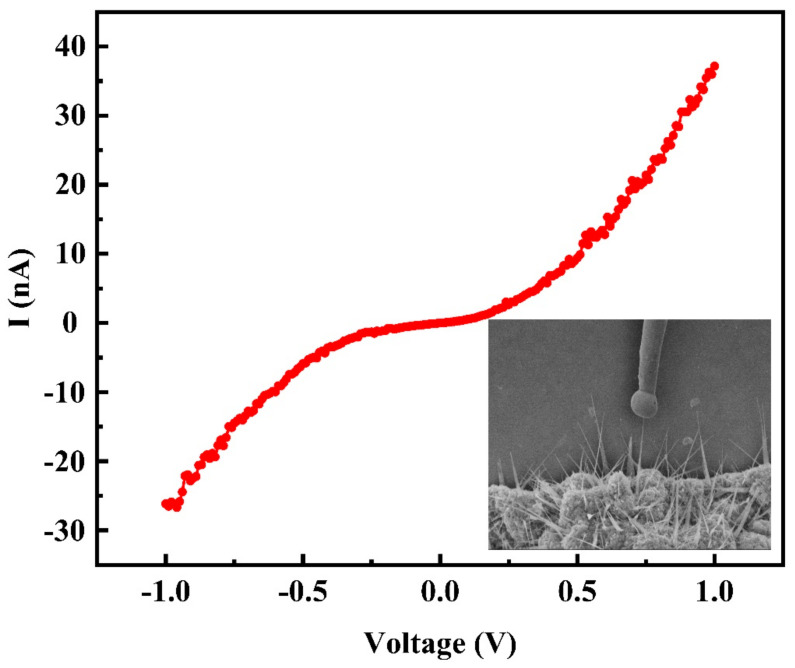
Typical electrical characteristics of an individual In-doped ZnO nanowire, inset is the SEM image during the measurement.

**Figure 9 nanomaterials-11-00240-f009:**
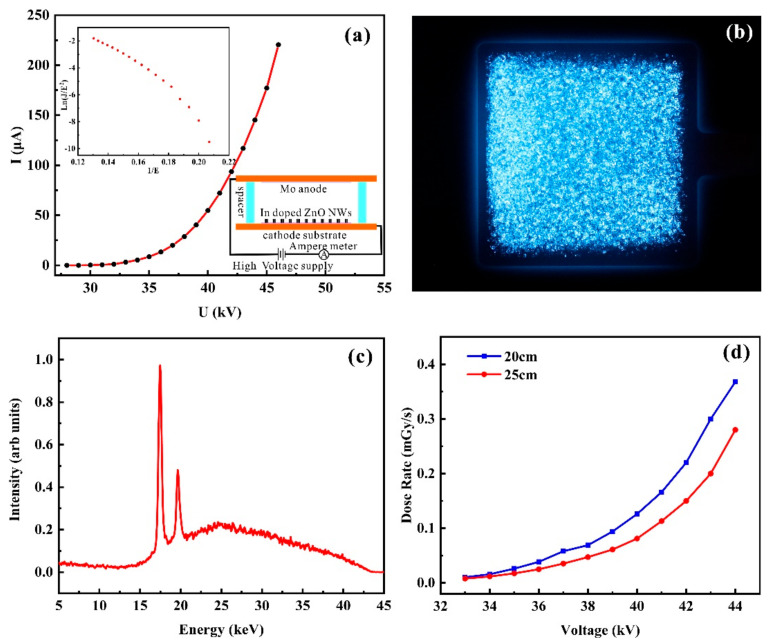
Characteristics of the flat-panel X-ray source. (**a**) I–V curve, (Upper Inset) F-N plot, and (Lower Inset) schematic diagram of testing device. (**b**) Visible light image recorded when the device is operated at a 46 kV anode voltage. (**c**) Normalized X-ray energy spectra at a 46 kV anode voltage. (**d**) X-ray dose rate vs. voltage for anode-cathode distance of 20 cm (blue squares) and 25 cm (red circles).

**Figure 10 nanomaterials-11-00240-f010:**
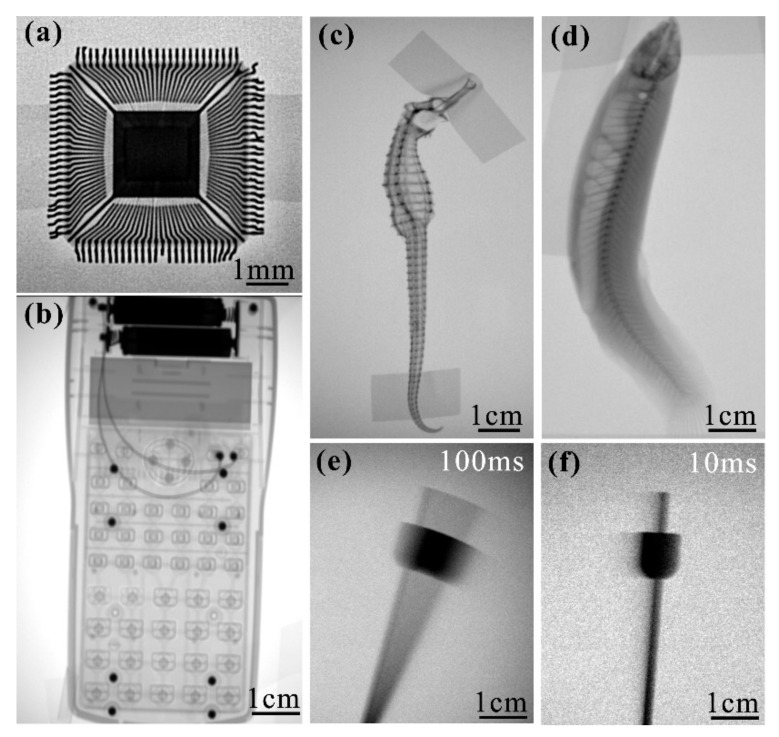
X-ray images of non-biological and biological samples (**a**–**d**). (**a**) An integrated circuit chip. (**b**) A calculator, (**c**) hippocampal specimen and (**d**) fresh loach. Images of moving metronome needle taken using the X-ray source driven using pulsed high voltage with the pulse width of 100 ms (**e**) and 10 ms (**f**).

**Table 1 nanomaterials-11-00240-t001:** Geometrical parameters, resistance and conductivity of single In-doped ZnO nanowire prepared using In-Zn film by thermal oxidation method.

Samples	Length (μm)	Diameter (nm)	Resistance (MΩ)	Conductivity (S/m)
1	3.8	80	35	22
2	4.4	41	182	18
3	5	40	56	71
4	4.5	83	54	15
5	5.1	65	8.8	174
6	5.5	69	16.7	88
7	7	57	65	42
8	4.4	67	7.2	173

## Data Availability

Data is contained within the article.

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
