# Peer review of "High Current Field Emission from Large-Area Indium Doped ZnO Nanowire Field Emitter Arrays for Flat-Panel X-ray Source Application"

_nanomaterials, 2021, doi:10.3390/nano11010240_

Round 1
Reviewer 1 Report
The authors are working on the synthesis of In-doped ZnO for field emission use, in real dig dimensions e.g. ~5cm2.
This is indeed an interesting topic that can find applications in flat-panel X-ray sources.
The authors follow the Thermal oxidation method which is indeed a feasible method to prepare large area samples. They used SEM, TEM, XRD, XPS and PL spectroscopy in order to characterize their samples.
Moreover the authors have extensively studied the growth mechanism of ZnO growth, while they have studied their field emission and their electrical properties using a probe technique.
I-V curves, F-N plots have been prepared, while typical X-ray images for non-biological and biological samples can be observed using the as-grown samples.
I feel that the present manuscript can be accepted for publication in MDPI Nanomaterials after minor revision.
A few typos, and some linguistic issues can be corrected.
Reviewer 2 Report
The authors have conducted a number of characterization and test for the field emission properties of In-doped ZnO NWs. There are severe concerns related to the interpretation of the characterization a data and the growth model.
[1] The Introduction Section is useful, surveying on the relevant literature. However, some information is repeated. Sight shortening would make this part more coherent.
[2] Figure 1(c): It seem that a not so uniform and dense growth of ZnO NWs takes place. Typical average thickness provided by SEM on a large number of NW should be provided.
[3] Only one In concentration has been considered here. Given that the authors has previously published a similar work on pure ZnO, it would be useful to provide information about a number of In doping concentration so as to correlate the field emission properties with this parameter.
[4] Starting from a Zn/In:80/20 ratio the final In doping level is of ~1%. The authors must provide a rationale for this outcome. Based on the spectroscopic data following, the 1% doping seems rather erroneous. XPS probes much larger sample areas and is hence more reliable in estimating the doping level than using a single NW with TEM.
[5] XRD data
(a) The authors do not provide the magnitude of the shift of the Bragg peaks arising from the in participation in the ZnO lattice.
(b) The observation of the not so weak 101 In peak in the XRD patter is dubious. Indium atoms are dispersed in the ZnO lattice and hence a Bragg peak originate from an In-type lattice is not reasonable.
(c) To account for the small shift of the XRD peaks the authors claim that: “In this process, the larger In 3+ ions partially substitute into the sites of the smaller Zn+2 and so the lattice constant increases”. This is a very naïve interpretation. Doping by 1% implies a rather large distance among In ions, which would not observably affect the ZnO Bragg peaks.
[6] XPS analysis
Binding energies should be given with one decimal point. The same for In peaks, which are mot given in the proper accuracy.
The interpretation of the XPS peaks with doping is very simplistic. More details are needed to explain the shift of the peaks binding energies.
A detailed estimation of In concentration must be provided by the XPS data. The authors should consider the sensitivity factors of each element and advance an elemental analysis. The XPS spectra reveal that the In doping should be much higher than 1% estimated by TEM analysis.
[7] PL data
The complete lack of a visible PL band in the In-doped crystal is in stark contrast with the very strong visible (defect) band in pure ZnO of the previous authors work, prepared by the same method. It is highly unlikely that the incorporation of Indium would completely suppress the defect band. On the contrary, doping is typically engendering the growth of the visible (defect) band in the PL spectra. The authors should perform again the experiment, following a statistical analysis by measuring many spectra from different samples to achieve a more reliable picture of the PL spectra.
[8] Raman
The Raman analysis is also incomplete. The authors should draw in the same figure the pure ZnO and the In:ZnO nanowires Raman spectra, in order to make easily visible to the reader the possible differences between them.
The feature at 67 cm-1 is not a true Raman band. It arises form the notch filter of the Raman set which block the elastic line below this spectra region.
There is a strong wide background below the main Raman bands (<600 cm-1). The authors should explain this feature. Also, the defect area is identified in the Raman spectra at ~580 cm-1. The presence of the background and the band at 580 cm-1 shows defect formation in In-doped ZnO NWs. This is in contrast to the PL data. More spectra should be recorded for a better statistical analysis of the Raman spectra.
[9] The description of the growth mechanism is very confusing. The initial film contains Zn and In in analogy 80 to 20. The authors state that Indium melts and particles precipitate onto the surface. This is a simplistic if not erroneous approach since the authors have overlooked the fact that they have to deal with an alloy characterized by a possibly eutectic melting point. The binary Zn-In phase diagram should provide the relevant information. Zn metal melts at 420 oC. Therefore, invoking vapors of Zinc is obviously a misleading concept. “Decomposition of indium particles into In atoms…” as stated by the authors is not an acceptable physical process. What is a bicrystalline structure?
The authors have to completely reconsider the proposed mechanism of nanowire growth.
[10] The phrase: “The F N pl of of In doped ZnO nanowires shows linear and non linearity at low electric fields and high electric fields respectively (inset of figure 6(a)” is misleading. The data show two distinct linear regimes which intersect at some point. The relevant discussion should be revised.
[11] The role of nanowire crystallinity on the field emission properties should be reconsidered, given the comments discussed above about the crystallinity difference between pure and doped ZnO nanowires.
Minor issues
(a) The inset axes and labels in figure 6(a) is not well visible.
(b) The authors state in the experimental part “a 1 mm thick molybdenum thin film on silica glass”. Is it really 1 mm thick or there is a typo here?
Reviewer 3 Report
The manuscript is devoted to preparation of large-area indium-doped ZnO nanowire arrays that show high field emission current by the thermal oxidation method. The growth process was explained using a self-catalytic vapor– liquid–solid growth mechanism while excellent electrical properties was ascribed to the high crystallinity and high conductivity induced by the indium dopants. The possible application of these In-doped ZnO nanowire arrays in a flat-panel X-ray source and corresponding imaging was demonstrated as well.
The subject is interesting from scientific and especially practical point of view
The manuscript is well organized, experimental methods are good chosen, results are clearly explained, discussion sounds convincing.
I found few type mistakes and have one suggestion.
Type mistakes
Pg. 1 Abstract
In this paper, In this paper…
Pg. 9 Results and discussion
without element doping ZnO nanowire (UV band peak at 394 nm) [27]. that is attributed
“.”
Pg. 12
“With increased growth duration, the indium particles become smaller and eventually are depleted Therefore,”
Suggestion
“without element doping ZnO nanowires grown” – maybe “un doped ZnO nanowires” or “intrinsic ZnO nanowires”
Reviewer 4 Report
I have read with extreme interest the paper from Zhao et al. reporting on the realization of field emitter from large area Indium doped ZnO nanowire arrays. The authors develop a feasible strategy for the synthesis of high quality ZnO nanowires onto indium-tin-oxide (ITO)-coated glass substrates and execute a thorough characterization of the materials, finding that the nanowires are indium doped. They characterize the field emission properties of the resulting device, obtaining a current density higher than 10 mA/cm2. I also particularly like the nice applications shown by the authors in employing the devices for X-ray characterization of some interesting samples.
Few points need to be carefully corrected before considering the paper suitable for publication:
- In the introduction, the authors state: “Among these methods, element doping is an effective way to enhance the emission current by changing the nanowire conductivity and the lattice structure.” I suggest the authors should insert some recent references to better explain the physical mechanism of this enhancement.
- The authors should not underestimate the geometrical effect of the ZnO nanowires in determining the field emission properties. Indeed, the field emission follows the Fowler–Nordheim equations (Proc. R. Soc. Lond. A, 1928, 119, 173–181). The question that arires is: could the tapered geometry of the ZnO nanowires (see the SEM images) induce the high efficiency of the field emitter? This point could be interesting enough even to look for in future works, also considering the control strategies for ZnO nanowires geometry by rational approaches (Nature Materials, 2011, 10, 596–601; Nano Energy, 2018, 46, 54-62).
- How was the doping indium concentration determined? From the EDX elemental mapping of the nanowires, it seems that the indium doping is discontinuous along the nanowires. Can the authors comment on that?
- “the origin of the strong peak at 67 nm in Figure 3(d) is unknown and requires further study.” I would suggest the authors to consider the possibility that this signal could arise from the random placement of the ZnO nanowires around each spot. This could in turn activate modes in the backscattering geometry (Solid State Sciences, 2009, 11, 865-869). I agree that this topic might be very interesting to consider in future works.
- There are many English grammar mistakes thorough the manuscript which can be easily corrected. For instance, in the abstract “In this paper,” is repeated two times.
Round 2
Reviewer 2 Report
The authors have tried to reply all comments and criticism raised in my first report. In some cases, the explanations can be considered rational; however, in most cases the authors failed to provide convincing answers.
Old comment [2]:
The authors present a useful statistical analysis of the lengths and diameters of the ZnO nanowires. My comment about “a not so uniform and dense growth of ZnO NWs” was meant to state that in each spot the ZnO NWs protrude radially form the bulky material in a sparse manner.
Old comment [3]:
The authors reply that “Unfortunately, no reliable doping was observed in the samples prepared using In-Zn alloys of In10/Zn90” is somehow a confession that the synthesis they did is not well-controlled and rather serendipitous. It is not explained why such a rich in Indium alloy (In10/Zn90) is not capable of provided a doped ZnO crystal. This fact undermines the merits and novelty of the current work.
Old comment [4]:
On thermodynamic grounds, there is absolutely no reason to expect a different Indium doping concentration in the ZnO NWs and the underneath ZnO film. The authors failed to provide a rationale about this point in the revised article. A statistical TEM analysis on a proper number of NWs must be done in case the doping concentration is finally decided via the TEM analysis.
Old comment [5]:
The authors provide here a confusing and self-contradictory reply. Being in a doping concentration of 1%, Indium is unlikely to experience accumulation into small In-O crystallites, let alone, to form such a crystal size that would be observable by XRD.
The authors then unreasonably state that the Indium amounts to 15% by XPS analysis. What is finally the concentration of the Indium that the authors accept as correct? The 1% or the 15%?
Old comment [6]:
See again comment [4]. There is no thermodynamic ground to support that the nanowires and the bulky ZnO have different In doping. The authors do not provide experimental verification for their different doping levels in Indium.
Old comment [7]:
The same comment as before applies here. The PL spectrum show in the reply letter is probably a more characteristic example of the grown samples. Anyway, even pure ZnO that the authors studied in an earlier paper showed similar PL spectrum. The complete absence of the visible PL band (defects) shown in the paper is suspicious.
Old comment [8]:
Removing the Raman spectra form the article hides the problem rather than resolving it. The authors should have placed in the same figure statistically reliable Raman spectra o the pure ZnO and the In:ZnO nanowires, because Raman spectra can furnish additional information on crystallinity and defects.
Old comment [9]:
The authors have not practically provided solid ideas to make the proposed model more plausible than it has been in the first submission. Even if evaporation may happen, the vapor pressure, is expected vanishingly low. What is the vapor pressure of gaseous Zn at the temperature of the experiment? The authors have to support their abstract ideas with experimental data if they want to keep this model in the paper.
Round 3
Reviewer 2 Report
--